# Assessment of the Nonlinear Electrophoretic Migration of Nanoparticles and Bacteriophages

**DOI:** 10.3390/mi15030369

**Published:** 2024-03-08

**Authors:** Adrian Lomeli-Martin, Zakia Azad, Julie A. Thomas, Blanca H. Lapizco-Encinas

**Affiliations:** 1Microscale Bioseparations Laboratory, Biomedical Engineering Department, Rochester Institute of Technology, 160 Lomb Memorial Drive, Rochester, NY 14623, USA; 2Thomas H. Gosnell School of Life Sciences, Rochester Institute of Technology, Rochester, NY 14623, USA

**Keywords:** bacteriophages, nanoparticles, nonlinear electrophoresis, microfluidics

## Abstract

Bacteriophage therapy presents a promising avenue for combating antibiotic-resistant bacterial infections. Yet, challenges exist, particularly, the lack of a straightforward purification pipeline suitable for widespread application to many phage types, as some phages are known to undergo significant titer loss when purified via current techniques. Electrokinetic methods offer a potential solution to this hurdle, with nonlinear electrophoresis emerging as a particularly appealing approach due to its ability to discern both the size and shape of the target phage particles. Presented herein is the electrokinetic characterization of the mobility of nonlinear electrophoresis for two phages (SPN3US and ϕKZ) and three types of polystyrene nanoparticles. The latter served as controls and were selected based on their sizes and surface charge magnitude. Particle tracking velocimetry experiments were conducted to characterize the mobility of all five particles included in this study. The results indicated that the selected nanoparticles effectively replicate the migration behavior of the two phages under electric fields. Further, it was found that there is a significant difference in the nonlinear electrophoretic response of phages and that of host cells, as first characterized in a previous report, illustrating that electrokinetic-based separations are feasible. The findings from this work are the first characterization of the behavior of phages under nonlinear electrophoresis effects and illustrate the potential for the development of electrokinetic-based phage purification techniques that could aid the advancement of bacteriophage therapy.

## 1. Introduction

Antibiotics are widely regarded as one of the most important medical developments of the 20th century [1]. The discovery of penicillin in 1928 marked the beginning of the antibiotic revolution, which transformed the way infectious diseases were treated worldwide [2]. In less than a hundred years since the discovery of penicillin, mankind is in grave danger of being faced yet again with a lack of effective treatment options for bacterial infections. This threat can be attributed to the emergence of antibiotic-resistant bacteria (ARB) as a consequence of the extensive misuse and overuse of existing antibiotics and the limited discovery of novel ones [2,3,4]. On top of having a terrible impact on human health, ARB-related infections have a US $20 billion societal annual cost, and if left alone this health crisis could lead to a 2.0–3.5% reduction from the gross domestic product annually by the year 2050 [5]. The threat of antibiotic resistance is real and, as such, there is an immediate need for the development of alternative and novel treatment strategies to combat bacterial infections. One such potential alternative treatment option, that predates the discovery and widespread availability of antibiotics, is the use of bacterial viruses, known as bacteriophages or phages, for the treatment of bacterial infections [6,7]. The earliest documented report of phages being used as a treatment for a bacteria-caused disease was in 1919 when the French physician Felix d’Herelle used phages to treat dysentery by means of an ingestible solution [6]. Since then, phage therapy was mostly practiced in a limited number of countries (e.g., Georgia and Poland) until the last few decades when the emergence of ARB has made phage therapy of interest globally [6,8,9,10].

This interest is supported by phages having a number of characteristics that make them an attractive option for the treatment of ARB infections, such as their high abundance in the environment and their genetic diversity [11,12]. Notably, their high genetic diversity results in different phages utilizing different mechanisms to infect and lyse a cell; thus, in instances of phage resistance by a bacterium, there would be other phages available that could be utilized for therapy instead [13]. In addition, since most phages only infect a relatively narrow spectrum of bacterial taxa (e.g., some phages can only infect a certain bacterial species) [14], they have little to no effect on the normal microflora of the human body [15,16]. For instance, the phages ϕKZ and SPN3US can only replicate on their isolation hosts, *Pseudomonas aeruginosa* and *Salmonella* Typhimurium, respectively; and cannot replicate on the other phage’s host, despite their diverged evolutionary history [17,18,19]. Both phages are sometimes referred to as “giant” or “jumbo” phages as they have genomes that are unusually long (>200 kbp) and highly complex virions compared to most other types of phages [20,21,22]. ϕKZ and related phages infecting *P. aeruginosa* have a long history of being used for treating infections caused by *P. aeruginosa* such as respiratory, surgical site, and urinary tract infections [17,23,24]. Similarly, SPN3US was isolated for its potential to control *Salmonella* in the food industry, along with many other giant phages targeting a broad spectrum of pathogens in recent years [18,25,26].

However, the usage of phages, including ϕKZ and SPN3US, as therapy options faces significant challenges before reaching widespread availability. One such challenge is the lack of a purification methodology that can be easily utilized for a wide diversity of phage types in a time-efficient manner. This is a direct consequence of the remarkable diversity of phage virions, in terms of structure, dimensions, and composition, as well as the variability in cellular debris that remains in phage samples after their growth from the host bacterium. Consequently, some purification methods simply cannot adequately purify phage virions from cellular debris to allow a phage sample to be utilized therapeutically [27,28]. Other approaches, including some known to produce highly purified phages, can damage the virions resulting in a significant loss of titer [28,29]. These issues mean that for any new phage type, a purification protocol has to be assessed for efficacy as even related phages may respond significantly differently to the same protocol [28]. In addition, current phage purification approaches require several steps, including the removal of the remaining cells and larger cell debris. Thus, phage purification can be time-consuming, potentially taking several days for some phages, a scenario that is far from ideal in terms of phage therapy, especially for critically ill patients.

Recently, microscale electrokinetic (EK) methodologies have seen notable progress in the identification, separation, and even purification of different types of microorganisms, including phages [30]. The most common EK phenomena include electroosmosis (EO), dielectrophoresis (DEP), and electrophoresis (EP) which can be linear (EP_L_) and nonlinear (EP_NL_). A recent surge of interest in EP_NL_ over the last decade has transformed the field of insulator-based EK (iEK). Specifically, the findings from Rouhi et al. [31], Tottori et al. [32], Cardenas-Benitez et al. [33], Bentor et al. [34], Ernst et al. [35], and Lomeli-Martin et al. [36] have all shed light on the impact EP_NL_ has on particle migration in microfluidic EK devices under DC and low-frequency AC electric fields. The first three reports [31,32,33] demonstrated that EP_NL_ has a major effect on iEK systems, capable of trapping and concentrating particles; whereas the three later reports [34,35,36] assessed the relationship between particle characteristics and their migration under EP_NL_. An attractive feature of EP_NL_ is that characteristics such as particle size and shape can be exploited to discriminate between target particles, which is not possible under solely EP_L_ effects [37,38]; thus, giving EP_NL_ important advantages over EP_L_. Hence, EP_NL_ represents a promising approach for separating phages from their bacterial hosts, as size and shape differences can be readily exploited.

Although knowledge of EP_NL_ effects is not new, with the earliest mentions in the early 1970s by Dukhin and collaborators [39,40], there are a surprisingly low number of experimental studies on the characterization of EP_NL_ [41]. This study presents the first experimental characterization of the migration of phages under the influence of EP_NL_ effects. A series of particle tracking velocimetry (PTV) experiments were performed employing the two distinct phages (SPN3US and ϕKZ) and three types of polystyrene nanoparticles that had size and surface charge characteristics similar to those of the phages. An additional objective of this work was assessing the potential of employing polystyrene nanoparticles as proxies for phages, to be used when testing new separation systems. The results illustrated that the selected nanoparticles behave similarly to the phages under EK forces, and thus, can be employed as proxies for the phages. The PTV experiments allowed for the characterization of the migration of all five particles (two phages and three nanoparticles) in this work under the effects of linear and nonlinear EP effects, which in turn resulted in the quantification of the mobilities of EP_L_ and EP_NL_ velocities. The dependence of the mobility of the EP_NL_ velocity, $\mu_{EP,NL}$, on particle size and shape was studied, where the size and shape of the phages were quantified employing the hydrodynamic diameter and the parameter of the sphericity [42], respectively. The results were consistent with previous reports, in which the magnitude of $\mu_{EP,NL}$ increases with particle size [35] and deviations from spherical shape [42,43], as indicated by lower sphericity values. Although the observed trend was subtle, it could be attributed to the limited size range (99–200 nm range) of the studied particles. A major outcome of this work is the identification of the significant difference between the values of $\mu_{EP,NL}$ of phages and those of bacterial host cells, which can enable future studies on the continuous separation between phages and host cells by exploiting these differences in $\mu_{EP,NL}$ values. These findings can contribute towards the development of novel phage purification protocols based on nonlinear EK phenomena.

## 2. Theory

Electrokinetic phenomena are classified according to their dependence on the electric field (**E**) as either linear or nonlinear. Linear EK phenomena, also called first-kind, have a linear dependence on **E** and are governed by the permanent surface charge. The linear EK phenomena considered here are EO and linear EP (EP_L_). Following the Helmholtz–Smoluchowski Equation, the EO velocity is defined as [44]:(1)$$v_{EO}=\mu_{EO}E=-\frac{\varepsilon_{m}\zeta_{W}}{\eta }E$$
where $\mu_{EO}$ is the EO mobility, $\varepsilon_{m}$ and η represent the media permittivity and viscosity, and $\zeta_{W}$ is the zeta potential of the channel wall. Regarding EP_L_, due to the small size of the nanoparticles and phages studied hereby, Henry’s formula [45] will be employed to estimate the mobility of EP_L_ ($\mu_{EP,L}$). A discussion on the selection of this expression is included in the Appendix A. The expression of the EP_L_ velocity is as follows:(2)$$v_{EP,L}=\mu_{EP,L}E=\frac{2\varepsilon_{m}\zeta_{P}}{3\eta }f(\kappa \alpha )E$$
(3)$$f\left(\kappa \alpha \right)=\left[1+\frac{1}{2{\left(1+\frac{2.5}{\kappa \alpha \left(1+2e^{-\kappa \alpha }\right)}\right)}^{3}}\right]$$
where $\zeta_{P}$ is the zeta potential of the particle, κ is the inverse of the Debye length ($\kappa =\lambda_{D}^{-1}$), and α is the radius of the particle. Further information on these expressions describing EP_L_ is included in the Appendix A.

Nonlinear EK phenomena, also called the second kind, have a nonlinear dependency with **E** and are a function of the bulk charge. This study considers nonlinear EP (EP_NL_), which in contrast to EP_L_, the mobility of EP_NL_ ($\mu_{EP,NL}$) does depend on the magnitude of **E**. Several models describe EP_NL_; these models are described by the dimensionless parameters of Peclet (*Pe*) and Dukhin (*Du*) numbers as well as the dimensionless field strength *β*. A description of these three dimensionless parameters is included in the Appendix A. Analytical expressions have been derived to describe the migration of particles under EP_NL_ in the limiting cases of small *Pe* (*Pe* ≪ 1) and large *Pe* (*Pe* ≫ 1); no analytical expressions exist for intermediate *Pe* values [46]. The expressions for these two limiting cases, illustrating the dependencies with **E**, for the velocity of EP_NL_ are [46] as follows:(4)$$v_{EP,NL}^{\left(3\right)}=\mu_{EP,NL}^{\left(3\right)}E^{3}\mathrm{for}\;\beta \leq 1,Pe<<1\mathrm{and}\;\mathrm{arbitrary}\;Du\;(\mathrm{moderate}\;\mathrm{field}\;\mathrm{regime})$$
(5)$$v_{EP,NL}^{\left(3/2\right)}=\mu_{EP,NL}^{\left(3/2\right)}E^{3/2}\mathrm{for}\;\beta >1,Pe>>1\mathrm{and}\;Du<<1\;(\mathrm{strong}\;\mathrm{field}\;\mathrm{regime})$$

Due to the small size of both the polystyrene nanoparticles and phages used in this work, *Pe* values above 1 were not reached with the current experimental conditions. Thus, for describing EP_NL_ effects, only the $\mu_{EP,NL}^{\left(3\right)}$ values, which correspond to the moderate field regime ($E^{3}$), will be reported. Details on the values of the *Pe* and *Du* numbers are included in Appendix A for both the weak ($E^{1}$) and moderate ($E^{3}$) electric regimes. With this, the overall velocity of a particle under the influence of an electric field in a post-less microchannel, such as the one shown in Figure 1, becomes the following:(6)$$v_{P}=v_{EO}+v_{EP,L}+v_{EP,NL}^{\left(3\right)}=\mu_{EO}E+\mu_{EP,L}E+\mu_{EP,NL}^{\left(3\right)}E^{3}$$

It is important to note that there are no dielectrophoretic effects present in this system since the electric field has a uniform distribution. The electrokinetic equilibrium condition (**E***_EEC_*, i.e., the electric field at which $v_{P}=0$) was also determined for each particle in this study. This parameter, proposed by Cardenas-Benitez et al. [33], can be used as an additional approach to estimate $\mu_{EP,NL}^{\left(3\right)}$ values for particles that are under the moderate regime when they reach$\;v_{P}=0$.

Since the phage capsids are non-spherical, and particle shape influences particle migration under EP_NL_ [42,43], it is necessary to quantify particle shape. Following a previous study from our group [42], the shape parameters of sphericity (ψ) was used to assess phage shape. The following is the expression for estimating ψ [47]: (7)$$\psi =\frac{\pi^{\frac{1}{3}}{\left(6V_{P}\right)}^{\frac{2}{3}}}{A_{P}}$$
where $V_{P}$ is the volume of the particle and $A_{P}$ is the surface area of the particle, sphericity varies from 0 to 1, where 1 means a perfect sphere.

## 3. Materials and Methods

Creation of microdevices. The microchannels used for the PTV experiments were made from polydimethylsiloxane (PDMS, Dow Corning, MI, USA) employing a standard cast-molding technique [48]. All microchannels, which featured no insulating posts, had the same dimensions: 10.16 mm in length, 0.88 mm in width, and 40 μm in depth. A schematic of the device employed for PTV experiments is shown in Figure 1.

Suspending medium. The suspending medium employed was 0.2 mM K_2_HPO_4_ solution with the addition of 0.05% (*v*/*v*) of Tween 20 to avoid particle adhesion to the device surface. This medium had a conductivity of 40.7 ± 4.0 µS/cm and a pH of 7.3 ± 0.2, which resulted in a wall zeta potential ($\zeta_{W}$) of −60.1 ± 3.7 mV and a $\mu_{EO}$ of 4.7 ± 0.3 × 10^−8^ m^2^ V^−1^ s^−1^ in the PDMS channels, as measured with current monitoring experiments [49]. In summary, the current monitoring experiment consisted of filling three parallel channel systems with the suspending medium. Platinum wire electrodes were placed at the reservoir tops, and 1000 V of DC voltage was applied to the channels. Initial stable current signals were recorded before changing the solutions in the reservoirs, applying the same potential, and recording the time response until the electric current reached a second plateau. A detailed description of this experimental procedure is included in the Appendix A and in the original publication of this methodology [49].

Nanoparticles. Three distinct types of fluorescent polystyrene nanoparticles (Magsphere Pasadena, CA, USA and ThermoFisher Scientific, Waltham, MA USA) of varying sizes and electrical charges were studied. Their properties are listed in Table 1. Nanoparticle samples were created by diluting the concentrated stock with the suspending medium. The concentration of each particle employed varied depending on size for optimum visualization and varied from 2.8 × 10^9^–9.0 × 10^10^ particles/mL as reported in Appendix A.

Viral samples. High titer stocks (10^10^–10^12^ pfu/mL) of two phages were employed in this study: SPN3US infective for *Salmonella enterica* Typhimurium LT2, and ϕKZ infective for *Pseudomonas aeruginosa*. To eliminate bacterial debris, all phage stocks underwent a low-speed centrifugation at approximately 8000 rpm for 10 min at 4 °C. The SPN3US samples were then fluorescently labeled using the following procedure: a 1 mL aliquot of phage stock was centrifuged at 13,000 rpm for 10 min. After discarding the supernatant, the resulting pellet was resuspended in 0.5 mL of distilled water. Next, 2 µL of SYTO 11 dye (Invitrogen, Carlsbad, CA, USA) was added to the sample and incubated for 20 min. Excess dye was then removed, and the sample was resuspended in 0.5 mL of the suspending medium. No dye was used for the ϕKZ samples. The properties of the characterized phages are listed in Table 2. The titers for each phage sample employed ranged from 8 × 10^11^–8 × 10^12^ pfu/mL as reported in Appendix A. Additional characteristics of the phages, including estimates of virion dimensions are shown in Appendix A. A discussion on how the hydrodynamic diameter (D_H_) of both phages was estimated, along with the equations used to do so, is included in the Appendix A.

Equipment and software. The LabSmith Sequencer software (V1.167) was used to control a high-voltage power supply (Model HVS6000D, LabSmith, Livermore, CA, USA) that applied constant DC voltage sequences to the microchannels using platinum-soldered electrodes. An inverted microscope was used to record the experimental runs: a Leica DMi8 (Wetzlar, Germany) microscope. 

Experimental procedure. To ensure a reproducible EO flow, the microchannel was conditioned with the suspending media for 8–12 h before experiments, and the liquid levels at both reservoirs were balanced to mitigate the effect that pressure-driven flow may have had on the system. A volume of 5–10 µL of the nanoparticle or phage sample, was introduced to the inlet reservoir, after which platinum wire electrodes were placed and fixed in both reservoirs. Nanoparticle and phage migration were observed and recorded at a range of applied voltages to observe both linear and nonlinear EK effects. Low-voltage PTV experiments in which the *Pe* value was below 1 were conducted to obtain $\zeta_{P}$ and$\;\mu_{EP,L}$ under conditions of the weak field regime (Table 1, Table 2 and Appendix A). High-voltage PTV experiments were subsequently conducted to acquire${\;\mu }_{EP,NL}$ under conditions of the moderate field regime (Table 1, Table 2 and Appendix A). For the estimation of $\mu_{EP,NL}^{\left(3\right)}$ values, velocity data obtained at electric field values that were the closest to the E_EEC_ value were employed. Additional estimations of $\mu_{EP,NL}^{\left(3\right)}$ values obtained by interpolating the E_EEC_ from velocity data are included in Appendix A. As expected, these results are similar to those in Table 1 and Table 2. All experiments were conducted in triplicate, and both the ImageJ and Tracker software (Version 5.1.5) were employed to determine particle velocity. A detailed description of the PTV experimental procedure and data analysis is included in the Appendix A.

## 4. Results and Discussion

### 4.1. Characterization of the Velocity Behavior of Polystyrene Nanoparticles and Bacteriophages

A series of PTV experiments were conducted by varying the magnitude of the electric fields (25–2400 V/cm) to determine the overall particle velocity of the three types of nanoparticles and the two phages studied in this work. The results are shown summarized in Table 1 and Figure 2. The electrophoretic velocity depicted in Appendix A, which considers the linear and nonlinear components, was obtained by subtracting the electroosmotic component from the overall particle migration. Appendix A illustrates that at high electric fields, the electrophoretic migration is no longer linear with the electric field, further supporting the presence of EP_NL_. The three nanoparticles were selected with two criteria in mind: (1) the nanoparticles must possess a diameter akin to the hydrodynamic diameter (D_H_) of the employed phages, and (2) the nanoparticles must possess a charge similar to that of the phages. Given the intermediary nature of the phages’ hydrodynamic and capsid diameters included in this study, these being ~150 nm, both 100 nm and 200 nm nanoparticles were studied. It was decided to utilize nanoparticles with an aminated surface functionalization since, from previous results in our laboratory [36], aminated particles possess electrical charges similar to those of microorganisms, which have a lower magnitude than the $\zeta_{W}$, allowing the particles to move forward, as represented in the inset in Figure 1. The selection of appropriate proxies for the phages is confirmed in Table 1. The $\zeta_{P}$ of the nanoparticles are similar to those of the phages and all have a magnitude below the $\zeta_{W}$ value of −60.1 mV.

From Figure 2a, which depicts particle velocity vs. electric field, it is seen that all three particles follow the expected behavior: a linear increase of their velocity at low electric fields, a maximum immediately followed by a decrease in velocity as the electric field increases, reaching negative velocity values. The two larger nanoparticles (Particles 2 and 3) crossed the zero-velocity threshold at a lower magnitude electric field, which is also an expected result since the effects of EP_NL_, the phenomena attributed to cause the decrease in velocity magnitude, increases at high electric fields and increases with particle size. The E_EEC_ values of the nanoparticles range between 1564.6 and 1710.5 V/cm. 

Regarding the phages, their migration behavior is illustrated in Figure 2b, confirming that the selected nanoparticles and the phages behave in a similar manner, even though the phages have slightly larger magnitudes in their $\zeta_{P}$ values. Furthermore, the phage ϕKZ crosses the zero-velocity line roughly 200 V/cm before SPN3US, whose genome length is smaller than that of ϕKZ (Appendix A). The *E_EEC_* values of the phages are 1640.6 and 1431.0 for the SPN3US and ϕKZ, respectively. It is noteworthy that the *E_EEC_* values for both nanoparticles and phages fall within a relatively narrow range of 1550 ± 150 V/cm. It is important to highlight that while both size and shape influence the EP_NL_ behavior of particles, the size and shape (in terms of sphericity values) of all nanoparticles and phages examined in this study are highly similar. In terms of their hydrodynamic diameter all particles range between 99 to 200 nm; while the sphericity values estimated for both phages are almost identical. Consequently, it is expected that all the nanoparticles and phages studied would exhibit very similar velocity behaviors, as evidenced in Figure 2.

### 4.2. Determination of the Mobility of Nonlinear Electrophoresis of Polystyrene Nanoparticles and Bacteriophages

The determination of the mobility of EP_NL_ is necessary for the design of EK-based separations. As was mentioned in the theory section, due to the minute dimensions of both the nanoparticles and phages, only the moderate electric field regime is reached, in which the EP_NL_ velocities have a cubic dependence on the electric field (Equation (4)). Previous studies have demonstrated that the magnitude of $\mu_{EP,NL}^{\left(3\right)}$ increases with increasing particle size [35,40] and increases with increasing deviations from spherical shape (decreasing values of sphericity, ψ) [42,43]. This was considered in Figure 3, which illustrates the $\mu_{EP,NL}^{\left(3\right)}$ magnitude as a function of the ratio of D_H_/ψ. According to previous experimental studies, the magnitude of $\mu_{EP,NL}^{\left(3\right)}$ should increase as a function of D_H_/ψ. This trend is only weakly observed. A potential cause of this slight variation of $\mu_{EP,NL}^{\left(3\right)}$ with D_H_/ψ could be the small overall size of all particles employed here, which diminishes the effect of particle size under this limited particle size range, since particle diameters only varied from 99–200 nm. Further, the overall differences in the estimated shape between the two phages is negligible, as illustrated by their almost identical sphericity values.

As observed in Figure 3, the $\mu_{EP,NL}^{\left(3\right)}$ magnitude of the ϕKZ is around ~30% higher than that of SPN3US, which could be considered unexpected as both phages have similar characteristics in size, shape, and zeta potential. There are two potential causes for the higher magnitude in $\mu_{EP,NL}^{\left(3\right)}$ for ϕKZ: (i) the longer genome length (~40 kb longer, Appendix A) of ϕKZ and (ii) the presence of hundreds of very thin and long tail fibers along the ϕKZ tail sheath, which are not present in the SPN3US phage. These fibers, called facultative structures, possess a narrow width and irregular structure [17,50], which makes it impossible to quantify their effect on the overall surface area (*A_P_*) of ϕKZ, which in turn would decrease the sphericity of ϕKZ (Equation (6)), displacing the data point corresponding to ϕKZ to the left in Figure 3, perhaps increasing its agreement with the slight trend depicted in Figure 3. These results add another interesting layer of complexity for EP_NL_ studies since they hint at the fact that particle morphology is also a determining factor for the values of $\mu_{EP,NL}^{\left(3\right)}$, or at the very least, it appears to be that way for particles in the nanometric scale. Further studies are required to verify this hypothesis. 

In terms of the overall magnitude of the $\mu_{EP,NL}^{\left(3\right)}$ values determined in this study for all five particles, a major outcome of this work is the significant difference between the $\mu_{EP,NL}^{\left(3\right)}$ values of nanoparticles and phages and those of bacterial host cells, as bacterial cells have $\mu_{EP,NL}^{\left(3\right)}$ with higher magnitudes [51]. A previous study from our group performed under similar conditions determined the $\mu_{EP,NL}^{\left(3\right)}$ of *Salmonella enterica* Typhimurium as −72.2 × 10^−19^ (m^4^V^−3^s^−1^), which is more than six times the magnitude of its corresponding phage, ϕKZ, that has a $\mu_{EP,NL}^{\left(3\right)}$ value of −11.7 × 10^−19^ (m^4^V^−3^s^−1^). This is highly encouraging, as it illustrates two important aspects: all nanoparticles employed here can effectively function as proxies for phages as they showed similar $\mu_{EP,NL}^{\left(3\right)}$ values to those of phages; and more importantly, there is great potential for separating phages from host cells in EK-based systems by exploiting differences in $\mu_{EP,NL}^{\left(3\right)}$ values.

## 5. Conclusions

This study presents the first report of the characterization of the $\;\mu_{EP,NL}^{\left(3\right)}$ of phages. A total of five particles were studied, three polystyrene nanoparticles and two phages, SPN3US and ϕKZ. The selection of three nanoparticles was meticulous to ensure that the nanoparticles had similar size and similar electrical charge to that of the phages. For the latter criteria, nanoparticles with an aminated charge proved to fulfill this requirement. The results illustrated a good match between the dielectric characteristics of selected nanoparticles and phages, both in their velocity behavior and the values of$\;\mu_{EP,NL}^{\left(3\right)}$. The results indicate that when proper criteria are set for selecting synthetic particles, these can act as excellent proxies for phages in EK studies. 

The results confirmed previous experimental studies on the effect of particle size and shape on the magnitude of $\mu_{EP,NL}^{\left(3\right)}$, which had shown that the magnitude of $\mu_{EP,NL}^{\left(3\right)}$ increased with particle size and with deviations from spherical shape (lower sphericity values). However, only a slight increasing trend was observed when plotting $\mu_{EP,NL}^{\left(3\right)}$ as a function of the ratio particle diameter/sphericity. This weak trend was attributed to the limited size range and small size of the particles studied in this work, as all diameters were in the 99–200 nm range. The results illustrated that the magnitude of the $\mu_{EP,NL}^{\left(3\right)}$ of ϕKZ was ~30% higher than that of SPN3US, even though both phages have similar diameters/sphericity ratios. The presence of hundreds of thin and long fibers, called facultative structures, along the ϕKZ tail sheath could be the reason for this difference, as SPN3US lacks these structures. These fibers on the tail of the ϕKZ phage, which are numerous and irregular, could increase the overall surface area of ϕKZ, which in turn, would lower its sphericity. However, due to the irregularity of the fibers, it is not possible to quantify their effect. This illustrates that morphology becomes a relevant parameter when characterizing microorganisms in the nanoscale range, such as phages. Further, these findings illustrated that distinct phages can differ substantially in their dielectric properties, opening the possibility for identification and sensing applications employing EK-based systems. 

A major outcome of this work is the identification of the significant difference between the values of $\mu_{EP,NL}^{\left(3\right)}$ of phages and those of bacterial host cells, as the host cells have much higher magnitudes of $\mu_{EP,NL}^{\left(3\right)}$, which can enable effective separation processes between phages and host cells. These findings show new possibilities for the development of a robust and comprehensive phage purification protocol based on nonlinear EK phenomena, which will be explored in future work.

## Figures and Tables

**Figure 1 micromachines-15-00369-f001:**
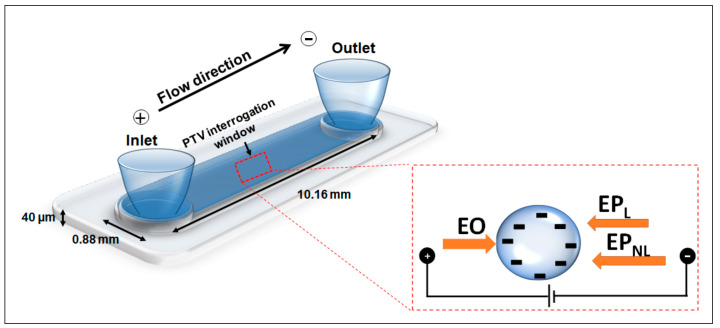
Schematic representation of the flat, post-less microchannel used for PTV experiments, including dimensions. The inset shows the considered EK forces along with their respective directions for a negatively charged particle and a channel with a negatively charged surface.

**Figure 2 micromachines-15-00369-f002:**
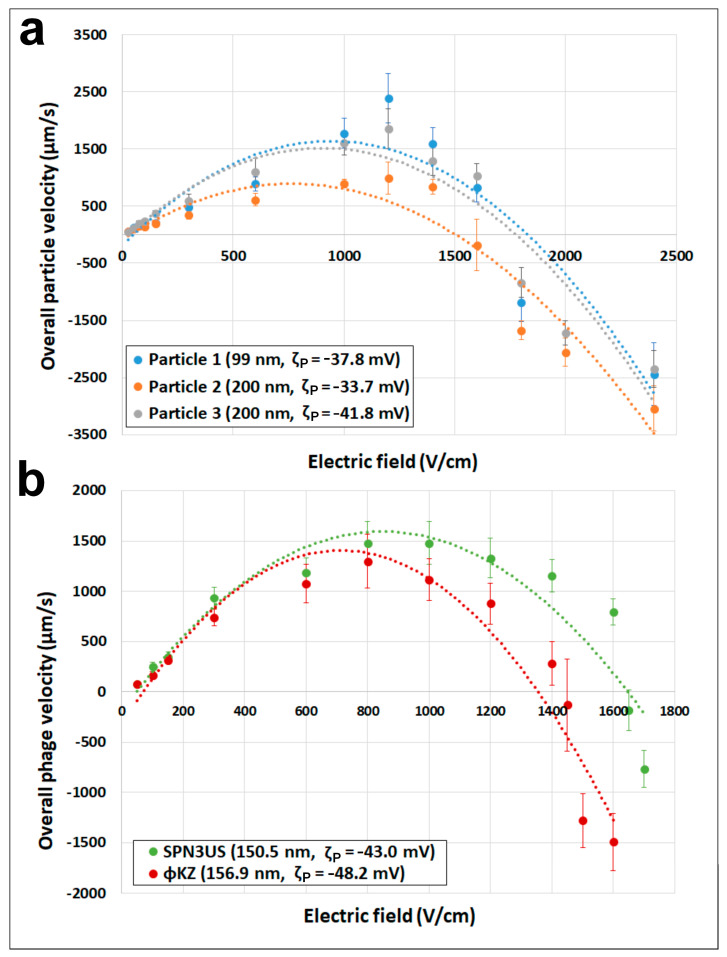
Overall velocity ($v_{P}$) as a function of the electric field (**E**) for (**a**) nanoparticles and (**b**) phages. Markers indicate experimental data, and the dashed lines are included for ease of visualization. Error bars denote standard deviation.

**Figure 3 micromachines-15-00369-f003:**
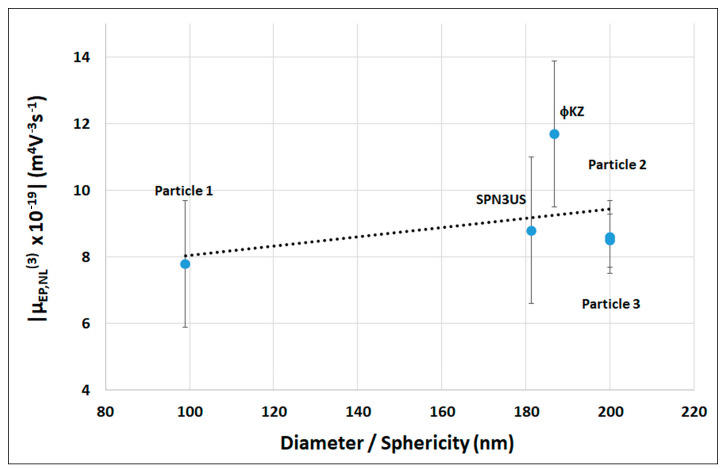
The absolute value of the mobility of nonlinear electrophoresis $\mu_{EP,NL}^{\left(3\right)}\;$ as a function of particle diameter divided by the particle sphericity. Absolute values were plotted for $\mu_{EP,NL}$ to aid visualization as these values are negative. For the phages, the hydrodynamic diameter was used. The sphericity of the spherical polystyrene nanoparticles was set to 1. Markers indicate experimental data, and the dashed line is included to denote the data trend, although the phage ϕKZ is outside the trend. Error bars denote standard deviation.

**Table 1 micromachines-15-00369-t001:** Characteristics of the particles employed in this study.

Particle ID	Diameter (nm)	$\zeta_{P}$ (mV)	$\mu_{EP,L}$ × 10^−8^ (m^2^ V^−1^ s^−1^)	$E_{EEC}$ (Vcm^−1^)	$\mu_{EP,NL}^{\left(3\right)}$ × 10^−19^ (m^4^ V^−3^ s^−1^)	$E\;\mathrm{for}\;\mu_{EP,NL}^{\left(3\right)}$ Determination
Particle 1	99 ± 18	−37.8 ± 1.1	−2.2 ± 0.1	1682.1 ± 61.1	−7.8 ± 1.9	1600
Particle 2	200 ± 20	−33.7 ± 1.6	−3.1 ± 0.3	1564.6 ± 83.9	−8.6 ± 1.1	1600
Particle 3	200 ± 12	−41.8 ± 6.9	−2.3 ± 0.1	1710.5 ± 33.7	−8.5 ± 2.5	1800

**Table 2 micromachines-15-00369-t002:** Characteristics of the phages employed in this study.

Phage Name	$D_{H}$ * (nm)	Sphericity	$\zeta_{P}$ (mV)	$\mu_{EP,L}$ × 10^−8^ (m^2^ V^−1^ s^−1^)	$E_{EEC}$ (Vcm^−1^)	$\mu_{EP,NL}^{\left(3\right)}$ × 10^−19^ ** (m^4^ V^−3^ s^−1^)	$E\;\mathrm{for}\;\mu_{EP,NL}^{\left(3\right)}$ Determination
SPN3US	150.5	0.83	−43.0 ± 8.1	−3.1 ± 0.2	1640.6 ± 49.8	−8.8 ± 2.2	1700
ϕKZ	156.9	0.84	−48.2 ± 4.8	−2.9 ± 0.3	1431.0 ± 42.1	−11.7 ± 2.2	1500

* The hydrodynamic diameter (*D_H_*) of the phages was estimated considering the entire volume of the phage (capsid and tail). Appendix A and the data in Appendix A were employed for the estimation of the *D_H_* values. ** Additional estimations of $\mu_{EP,NL}^{\left(3\right)}$ were performed at the E_EEC_ condition. These results, which are similar to the values in this table, are included in Appendix A.

## Data Availability

Data are contained within the article and Appendix A.

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
