# Peer review of "Assessment of the Nonlinear Electrophoretic Migration of Nanoparticles and Bacteriophages"

_micromachines, 2024, doi:10.3390/mi15030369_

Round 1
Reviewer 1 Report
Comments and Suggestions for Authors
The authors have extensive experience in separating particles using linear and nonlinear electrodynamics, and this article applies this technique to the isolation of bacteriophages at the submicron scale. This is new and interesting. In the article, the parameter of 𝜇𝐸𝑃,𝑁𝐿 (3) was proposed to identify the significant difference between phages and bacterial host cells. It can be published in micromachines after revision.
Comment:
1) In the introduction, the clinical significance of the two phages should be explained.
2) The experiment to separate the phages and host cells/cell had better be added to verify the effectiveness for the bacteriophage purification.
Reviewer 2 Report
Comments and Suggestions for Authors
The manuscript provides an analysis of the nonlinear electrophoretic migration of nanoparticles and bacteriophages. It is well-crafted, articulate, and merits publication. Nonetheless, some areas require clarification and improvement:
1. Fig. 1 is ambiguous. The nature of the circle in the "zoomed-in" segment is unclear. It is presumed to represent a "post" from the authors' prior research, but this is not immediately evident without consulting previous works and the methodological section. A clearer depiction is necessary.
2. The manuscript should specify the number of "posts," their arrangement, and their dimensions directly, rather than referring readers to previous publications for this fundamental information.
3. Clarification is needed regarding the data in "Figure S1. Electrophoretic velocity…" It appears to be derived from a "no posts" area within the microfluidic channel. If "overall velocity of a particle" is denoted as Vp (equation 6), the distinction between Fig. 2 and Fig. S1 requires explanation, as the text above Fig. S1 suggests otherwise.
4. The methodology for determining electroosmotic flow should be summarized in this manuscript. It can be found from a 2016 publication but the reader has to look for it. This is crucial as it is subtracted from Vp to obtain the Vl+Vnl component.
5. Similarly, the Particle Tracking Velocimetry (PTV) experiments need a description of the experimental procedure, data analysis, and how particle velocities are calculated. This detail is essential for enabling the replication of the study based solely on the information provided in the manuscript.
